# Teletherapy and hospitalizations in patients with serious mental illness during the COVID-19 pandemic: A retrospective multicenter study

**Antonio José Sánchez-Guarnido**[1], **Beatriz Machado Urquiza**[1], **Maria del Mar Soler Sánchez**[2], **Carmen Masferrer**[3], **Francisca Perles**[4], **Eleni Petkari**[5]*

1 Hospital Santa Ana, Motril, Spain, 2 Hospital Psiquiátrico José Germain de Leganés, Leganés, Spain, 3 Centre Fòrum de l´Hospital del Mar, Institut de Neuropsiquiatria i Addiccions, Parc de Salut Mar, Barcelona, Spain, 4 Hospital de la Axarquía, Málaga, Spain, 5 Universidad Internacional de la Rioja, La Rioja, Spain

* eleni.petkari@unir.net

## Abstract

### Background

Interventions with patients with Serious Mental Illness (SMI) had to adapt rapidly to the COVID-19 safety restrictive measures, leading to the widespread use of teletherapy as an alternative.

### Objectives

The aims of this study were to compare the use of different intervention modalities with patients with SMI during the first wave of the pandemic and examine their association with emergency hospital visits and hospitalization rates six months later.

### Methods

Records of 270 service users of fifteen outpatient mental health services across Spain, were retrospectively assessed. We retrieved clinical data and data on the modality of intervention received (in-person, over the phone, videoconferencing) in three time points (before, during and after the first COVID-19 wave). Also, data were retrieved regarding the frequency of their emergency hospital visits and hospitalization rates, two, four and six months later.

### Results

During the first wave of the pandemic, teletherapy (over the phone and videoconferencing) was the modality most widely used, whilst in-person therapy sessions were significantly reduced, though they seemed to return to pre-COVID levels after the first wave had passed. Importantly, patients receiving teletherapy during the first wave seemed to have significantly fewer emergency visits and hospitalization rates four and six months later ($\chi^2$ = 13.064; $p$ < .001). Multilevel analyses revealed that patients receiving videoconferencing interventions had fewer hospitalizations six months after the first wave (OR = 0.25; $p$ = .012).

**Data Availability Statement:** Our data cannot be shared publicly because they contain sensitive patient information, such as sex, age, living and

occupational status, and most importantly, diagnosis and records of mental healthcare visits. Such information is based on the clinical records of specific community healthcare centers, where most of the authors of this paper work as clinicians. Therefore, we consider that there is a risk of potential identification of the patients. To avoid such risk, access to the database can be formally requested through contacting the Biomedical Research Ethics Committee of the Government of Andalusia, Spain in the following address portaldeetica.csalud@juntadeandalucia.es.

**Funding:** AS received funding by the FPS 2020 – Primary Care Regional Hospitals and CHARES R&D Projects, (within the Project AP-0028-2020-C1-F2) and AS, MS, CM and FP received funding by the Outpatient Mental Health Day Hospitals of the Spanish Association of Neuropsychiatry. The funders had no role in study design, data collection and analysis, decision to publish, or preparation of the manuscript.

**Competing interests:** The authors have declared that no competing interests exist.

## Conclusions

Under challenging circumstances as those created by the COVID pandemic, teletherapy is a useful tool for protection against hospitalizations and can be used as an alternative to in-person therapy, to ensure continuity of care for patients with SMI.

## Introduction

The impact of the COVID-19 pandemic [1] on physical and mental health [2] has been widely reported in the literature. Social distancing, uncertainty about the crisis caused by the pandemic, and the consequent economic collapse, may negatively affect mental health, particularly in individuals with pre-existing mental disorders [3, 4]. Under such circumstances, individuals with Serious Mental Illness (SMI) have been shown to experience greater anxiety than people without previously reported psychopathology [5].

Social distancing practices may pose a great negative impact on individuals with psychotic disorders, as they are known to have small and low quality social networks [6], and are under increased risk of suicide due to isolation [7]. Furthermore, their ability to cover their basic needs may also be compromised, as many of them rely on community services (i.e. employment, occupational, and mutual support groups) that are less accessible under the circumstances created by the pandemic [8].

At the same time, the health crisis caused by the COVID-19 pandemic has posed a major challenge to mental health services. At the peaks of the pandemic, in-person interventions were limited, as they were considered a risk regarding coronavirus transmission. However, since continuity of care plays a key role in preventing the exacerbation of symptoms, and in reducing emergency consultations, hospital admissions, and suicide attempts [9], it was considered vital to find alternative ways of maintaining that continuity without exposing people with SMI to a greater risk of infection.

Alternatives to in-person care were in fact implemented with the aid of teletherapy, un umbrella term for what are known as telematic psychological interventions (telepsychotherapy and e-therapy) [10, 11]. Teletherapy is based on information and communications technology (computer-based Internet tools, mobile and land telephone calls, emails, fax, text messages, and videoconferencing consisting of patient-clinician communication through video consultations) [12]. It is a feasible and effective therapy modality that does not inhibit the therapeutic relationship [13–15] and seems to be widely accepted by both patients and professionals [16]. Despite some reported barriers associated with fears regarding loss of confidentiality, privacy, and legal regulations [14, 15], teletherapy seems to be a valid alternative for treating common mental health disorders, such as anxiety, depression, post-traumatic stress, and eating disorders [17, 18]. Videoconferencing, for example, has been found to be efficient with patients suffering from a variety of mental disorders [19] and is considered a useful tool for maintaining key aspects of the therapeutic process, such as therapeutic alliance [20]. Similarly, help lines are useful when patients do not have an adequate social network [21], while over the phone interventions are effective in reducing depressive symptoms [22, 23]. Teletherapy has been successfully used to implement several treatment models, including cognitive behavioral therapy [24], dialectical behavioral therapy [25], interpersonal therapy [22], and psychoanalysis [26], among others.

But despite the reported benefits of its use with individuals suffering from different mental disorders [19, 27], it has been barely explored in individuals with SMI, and even those studies

that have been carried out rarely included patients with increased symptom severity [28]. However, the provision of therapy through telephone calls, Internet, and videoconferencing does, however, seem feasible in patients with schizophrenia and related conditions [29]. The results of a recent systematic review [30] comparing the effectiveness of in-person and video-conferencing interventions in patients with psychosis showed similar levels of effectiveness in terms of health, psychiatric symptoms, and functionality. Also, in two of the studies included in this review treatment adherence was greater when videoconferencing was used. Overall, patients displayed high levels of satisfaction with teletherapy, this modality always being pre-ferred to the waiting list, or having to travel a long distance to the hospital.

One of the reasons behind the limited use of teletherapy with patients with SMI and the lack of research in this regard may be that teletherapy is frequently studied as a complement to treatment as usual [31], and thus, its efficacy as the primary therapeutic modality is not clearly recognized [28]. Also, most studies are characterized by methodological shortcomings, such as small sample sizes, absence of control groups, lack of randomized clinical trials, or short fol-low-up periods [27, 30]. Importantly, there may also be some sample selection bias, with the inclusion of patients and therapists who have more favorable attitudes towards this interven-tion modality [28]. Moreover, the use of digital technology among individuals with SMI is lower compared to the general population [32].

However, the COVID-19 pandemic has forced a transition from in-person therapy to tele-therapy in patients with more severe conditions, even among patients and health professionals whose attitudes that were not so positive towards this intervention modality [17]. To date, no studies have been carried out into teletherapy outcomes with patients with SMI under adverse psychosocial and health circumstances, such as those created by the COVID-19 pandemic. During the SARS epidemic in 2003, smartphones and internet services were not widely avail-able, and online mental health services were scarce [33], so the study of teletherapy applica-tions was not considered relevant.

It is therefore necessary for research into the impact of COVID-19 to focus on the provision of mental health care for the most vulnerable groups, such as the individuals with SMI. Many of these patients experienced interruptions to the continuity of their care, due to the measures taken to reduce infection rates over the first few months of the pandemic, while others man-aged to maintain contact with their services through teletherapy. To the best of our knowledge, to date no studies have examined how services were provided to such patients, and how the differences may be associated with a risk of hospitalization.

Therefore, the present study aimed to: a) explore the types of care offered to people with SMI during the first COVID-19 wave, the alternatives used when in person interventions were not possible, and the changes in the modality of interventions used over time (before the first wave, during lockdown, and after the first wave); b) examine whether receiving teletherapy, compared to not receiving teletherapy during the lockdown was associated with the frequency of visits to the emergency department and of hospital admissions, two, four, and six months after the lockdown; and c) examine if different teletherapy modalities are associated with hos-pitalization rates six months after the lockdown.

## Materials and methods

The Strobe checklist was used to prepare the manuscript (See S1 Checklist).

### Study design

Retrospective multicenter cohort study.

## Setting and participants

The Spanish mental healthcare system

Mental healthcare provision in Spain is quite complex. Among other facilities it includes a) the community mental health units, which constitute the first level of care units for people with SMI and provide both outpatient and homecare service; b) the mental health day hospitals, which constitute an intermediate resource between community mental health and hospitalization units, and provide specialized care in an outpatient/midday-stay mode; c) the inpatient units in general hospitals, providing full and partial hospitalization. Emergencies are handled by the general hospital emergency departments (psychiatry section), as well as the outpatient mental health units. For further information see: National Health System [34]. Fifteen community mental health hospitals in Spain were selected using stratified sampling, to achieve generalization in the Spanish population of people with SMI. Serious Mental Illness (SMI) was defined based on the ICD diagnosis, and on the intensity of the required care required [35]. This criterion was implemented by selecting all the patients who, due to the severity of their condition required follow-ups in an outpatient mental health hospital during the time period of the study. As this is a retrospective study based on clinical records, only patients who fulfilled the above criteria were included.

Patients with incomplete data were excluded from the study.

The total sample consisted of 270 people with SMI over 18 years old of age, nearly reaching the minimum required sample size of (N = 272) as calculated using the GPower analysis [36]. To calculate the sample size, we considered a 20% relapse rate for the group that received teletherapy and 30% relapse rate for the group that did not, with a potential sample loss of 15%, a Confidence Interval of 95% and 80% statistical power (see S1 File).

## Variables and data sources/measurement

Three two-month observation periods were established for the first wave of COVID-19 (2020): the period before the pandemic (January 16 to March 15), the lockdown period (March 16 to May 15), and the period following the first wave (May 16 to July 15).

The following variables were collected based on the patients' clinical records:

*Sociodemographic variables*: sex, age (in years), living status, occupational status, maximum level of education attained.

*Clinical variables*: Diagnosis based on the ICD-10 classification [37].

Treatment Adherence (yes/partially/no).

Treatment adherence was determined using the Medication Possession Ratio, defined as the proportion of time when medication supply is available [38].

Use of psychological interventions (yes/no): in-person (individual/group), over the phone, videoconferencing (individual/group)

Modality of intervention received (in-person/teletherapy).

Type of teletherapy received (videoconferencing individual/group, over the phone)

**Intervention characteristics.** In-person: once per week, individual (45–60 minutes)/ group (90 minutes)

Over the phone: once per week, individual (30–45 minutes). Sessions were held using the health service's landlines, following the same procedures as routine calls.

Videoconferencing: once per week, individual (30–45 minutes, one to one basis)/group (90 minutes, six to eight participants). The sessions were held through the health system enterprise video-calls software and were similar across sites.

*Outcome variables*: assessed two, four, and six months after the end of the first wave.

Percentage of hospitalizations (defined as admissions to a Mental Health Hospitalization Unit) calculated against the study sample

Mean number of visits to a Mental Health Emergency Department.

## Procedure

After receiving the approval of the corresponding ethics committees of the participating centers, data were collected retrospectively (October to November 2020) based on the patients' clinical records. The overall approval was obtained by the Medical Research Ethics Committee of the Andalusian Government stating the following: "This study fulfills the ethical principles required for conducting studies of this type" (REF: 202077133825). Patients were informed verbally and in writing about the project aims, and signed an informed consent. A password-protected database with sound error prevention mechanisms was designed, which granted access only to the study researchers; clinical data were handled without patient identification details. The study was performed in line with the principles of the Declaration of Helsinki. Data privacy requirements were met in accordance with the European Union legislation.

## Statistical methods

Statistical analyses were performed using the SPSS software v.21.0 [39], with a statistical significance of p-values < .05. We first examined the sociodemographic and clinical characteristics of our sample with descriptive statistics.

To study the first aim, we used a series of Cochrane's tests to determine whether there were statistically significant differences in the proportion of patients making use of the different intervention types over the three time points (before, during, and after the lockdown). We also performed pairwise comparisons through a series of McNemar's tests, to check whether the proportion of patients using the interventions was sustained or varied from one time point to another (before, during and after the lockdown). For these analyses, we applied a Bonferroni correction to the significance levels as follows: p = .005/3 = .016. For the second aim, we used Chi-square tests to compare the hospitalization rates at two, four, and six months after the first wave, and Student's t-tests were used to compare the mean number of visits to the Mental Health Emergency Department, between the patients who received teletherapy during the lockdown and those that did not. For the third aim, we first used Chi-square tests to compare the hospitalization rates six months after the lockdown between the patients that received each type of teletherapy (over the phone, videoconferencing individual and group) and those that did not. We then performed a mixed effects multilevel logistic regression including random effects for each patient nested in the three time points (before, during and after the first wave), and adjusting for individual characteristics, interdependence effects and confounding variables. We established Level 1 for interventions received and level 2 for the individual characteristics of the patients nested in them. The dependent variable was the frequency of hospitalizations 6 months after the first wave. In the first step the null model resulted in an ICC = 0.36, explained by level 2 variables, confirming the consideration of multilevel models. The first model included level 1 variables: videoconferencing (individual and group were considered together, due to lack of sufficient cases), over the phone, and in-person. For the second model we added the level 2 sociodemographic variables (sex and age) and for the third model we added the level 2 clinical variables (diagnosis and treatment adherence).

## Results

### Participants and descriptive data

Our sample consisted of 120 men and 150 women (55.6%), aged between 18 and 67 years (M = 39.90 years). Most of the participants had primary (35.8%) or secondary (41.5%) education, and 14.8% had university studies. The majority of the patients lived with their family of origin (28.9%), their own family (28.9%), or alone (17%); 29.3% were retired, 26.3% were unemployed, 20% had short-term disability, and 16.7% were working (Table 1). The most

**Table 1. Sociodemographic and clinical characteristics of the sample.**

| | Total *N* (%) | Patients who received teletherapy *N* (%) | Patients who did not receive teletherapy *N* (%) | *t/χ²* p-value |
|---|---|---|---|---|
| **Total Sample *N*** | 270 (100%) | 175 (64.8%) | 95 (35.2%) | |
| **Age *Mean (SD)*** | 39.90 (11.81) | 39.93 (11.12) | 39.84 (13.06) | *t* = -0.059 |
| | | | | *p* = .953 |
| **Gender** | | | | *χ²* = 0.208 |
| Female | 150 (55.6%) | 99 (56.6%) | 51 (53.7%) | *p* = .649 |
| Male | 120 (44.4%) | 76 (43.4%) | 44 (46.3%) | |
| **Living status** | | | | *χ²* = 5.664 |
| Family of origin (parents w/wo siblings) | 78 (28.9%) | 52 (29.7%) | 26 (27.4%) | *p* = .462 |
| Own family (partner and/or children) | 78 (28.9%) | 54 (30.9%) | 24 (25.3%) | |
| With friends or siblings | 16 (5.9%) | 8 (4.6%) | 8 (8.4%) | |
| One parent w/wo siblings | 37 (13.7%) | 23 (13.1%) | 14 (14.7%) | |
| Other | 7 (2.6%) | 6 (3.4%) | 1 (1.1%) | |
| Single household | 46 (17%) | 26 (14.9%) | 20 (21.1%) | |
| Supported housing | 8 (3%) | 6 (3.4%) | 2 (2.1%) | |
| **Occupational status** | | | | *χ²* = 6.286 |
| Student | 20 (7.4%) | 13 (7.4%) | 7 (7.4%) | *p* = .279 |
| STD | 54 (20%) | 39 (22.3%) | 15 (15.8%) | |
| Retired, pensioner | 79 (29.3%) | 48 (27.4%) | 31 (32.6%) | |
| Unemployed | 71 (26.3%) | 50 (28.6%) | 21 (22.1%) | |
| Working | 45 (16.7%) | 24 (13.7%) | 21 (22.1%) | |
| Volunteer/Mutual support agent | 1 (0.4%) | 1 (0.6%) | 0 (0.0%) | |
| **Level of Education** | | | | *χ²* = 2.993 |
| No schooling | 8 (3%) | 3 (1.7%) | 5 (5.3%) | *p* = .559 |
| Primary (BGE-CSE) | 96 (35.6%) | 61 (34.9%) | 35 (36.8%) | |
| Secondary education | 112(41.5%) | 76 (43.4%) | 36 (37.9%) | |
| University studies (BSc) | 40 (14.8%) | 26 (14.9%) | 14 (14.7%) | |
| Postgraduate studies (MSc-PhD) | 14 (5.2%) | 9 (5.1%) | 5 (5.3%) | |
| **Diagnosis (ICD)** | | | | *χ²* = 5.743 |
| Schizophrenia/Other psychosis | 82 (30.4%) | 50 (28.6%) | 32 (33.7%) | *p* = .570 |
| Bipolar disorder | 28 (10.4%) | 18 (10.3%) | 10 (10.5%) | |
| Personality disorders | 75 (27.8%) | 53 (30.3%) | 22 (23.2%) | |
| Depressive disorder | 26 (9.6%) | 17 (9.7%) | 9 (9.5%) | |
| Anxiety/Other disorders | 59 (21.9%) | 37 (21.1%) | 22 (23.1%) | |
| **Treatment Adherence** | | | | *χ²* = 0.422 |
| Yes | 232(85.9%) | 152(86.9%) | 80 (84.2%) | *p* = .810 |
| Partially | 27 (10%) | 16 (9.1%) | 11 (11.6%) | |
| No | 11 (4.1%) | 7 (4%) | 4 (4.2%) | |

frequent diagnosis was Schizophrenia or other psychosis (30.4%), followed by personality (27.8%), bipolar (10.4%), or major depressive disorders (9.8%). The majority showed treatment adherence (85.9%). No statistically significant differences were found between the patients that received some type of teletherapy and those who did not, in terms of sex ($\chi^2$ = 0.208; $p$ = .649), age ($t$ = -0.059; $p$ = .953), occupational status ($\chi^2$ = 6.286; $p$ = .279), level of education ($\chi^2$ = 2.993; $p$ = .559) diagnosis ($\chi^2$ = 3.853; $p$ = .426), or treatment adherence ($\chi^2$ = 0.422; $p$ = .810).

## Outcome data and main results

**Comparison between the different types of intervention before the pandemic, during lockdown, and after the first wave.** Table 2 displays the percentages of patients who received the different types of interventions over the three time periods. Overall, there were statistically significant differences across the usage of all types of interventions, as indicated by the Cochrane's Q results (Table 2). In-person individual interventions decreased from 80.4% before the pandemic to 23% during the lockdown ($p$ < .001), and then increased to 73.7% after the first wave ($p$ < .001); there were no significant differences in the use of in-person individual interventions after the first wave, in comparison with before the pandemic ($p$ = .043).

In-person group interventions decreased from 29,3% before the pandemic to 1,1% during the lockdown ($p$< .001) and then increased to 23,3% after the first wave ($p$< .001). There were no significant differences in the use of in-person group interventions after the first wave, in comparison with before the lockdown ($p$ = .034).

The percentage of patients who received over the phone interventions increased from 4.4% before lockdown, to 60% during the lockdown ($p$ < .001), and then significantly decreased to 21.9% after the first wave ($p$ < .001). However, this percentage was still significantly higher than before the lockdown (p < .001).

The percentage of patients receiving an intervention through individual videoconferencing before the pandemic increased from 0% to 3.7% during the lockdown ($p$ < .001) and decreased, although not significantly, to 1.5% after the first wave ($p$ = .146). The differences between the period before and after the first wave ($p$ < .125) were also not statistically significant.

Lastly, patients receiving videoconferencing group interventions increased from 0% before the pandemic to 9,3% during the lockdown ($p$ < .001) and to 11,1% after the first wave ($p$<

**Table 2. Comparison of interventions received before the pandemic, during the lockdown, and after the first wave.**

| Type of intervention | Before the lockdown (Jan 16 –March 15) % | During the lockdown (March 16 –May 15) % | After the lockdown (May 16 –July 15) % | Cochrane's Q | $\chi^2$ /p values | | |
|---|---|---|---|---|---|---|---|
| | | | | | Before vs during the lockdown | Before vs after the lockdown | During vs after the lockdown |
| **In-person individual** | 80.4 | 23 | 73.7 | Q = 224.573 p < .001 | $\chi^2$ = 142.012 p < .001 | $\chi^2$ = 4.129 p = .042 | $\chi^2$ = 125.823 p < .001 |
| **In-person group** | 29.3 | 1.1 | 23.3 | Q = 99.299 p < .01 | $\chi^2$ = 70.313 p< .001 | $\chi^2$ = 4.500 p = .034 | $\chi^2$ = 54.391 p< .001 |
| **Over the phone** | 4.4 | 60 | 21.9 | Q = 214.048 p> .001 | $\chi^2$ = 144.162 p < .001 | $\chi^2$ = 38.473 p < .001 | $\chi^2$ = 85.983 p < .001 |
| **Videoconferencing individual** | 0 | 3.7 | 1.5 | Q = 11.692 p = .003 | p = .002 | p = .125 | p = .146 |
| **Videoconferencing group** | 0 | 9.3 | 11.1 | Q = 44.286 p < .001 | p < .001 | $\chi^2$ = 28.033 p< .001 | p = .302 |

**Table 3. Hospitalizations and emergency visits two, four and six months after the lock down.**

| | Total (%) | Patients who received teletherapy (%) | Patients who did not receive teletherapy (%) | RR | 95%CI | $\chi^2$ /p-value |
|---|---|---|---|---|---|---|
| **Hospitalizations** | | | | | | |
| Two months after the lockdown | 6.7 | 4.6 | 10.5 | 0.43 | 0.18 to 1.06 | $\chi^2$ = 3.334 p = .068 |
| Four months after the lockdown | 13.7 | 9.1 | 22.1 | 0.41 | 0.23 to 0.75 | $\chi^2$ = 8.350 p = .004 |
| Six months after the lockdown | 20.4 | 13.7 | 32.6 | 0.42 | 0.26 to 0.67 | $\chi^2$ = 13.064 p< .001 |
| **Number of visits to the Mental Health Emergency Department** | | *Mean (SD)* | *Mean (SD)* | *MD* | | *t /p-value* |
| Two months after the lockdown | | 0.22 (0.80) | 0.55 (1.64) | 0.33 | 0.03–0.62 | t = 2.185 p = .030 |
| Four months after the lockdown | | 0.42(1.28) | 1.18 (3.52) | 0.76 | 0.18–1.35 | t = 2.571 p = .110 |
| Six months after the lockdown | | 0.70 (2.19) | 1.65 (5.14) | 0.95 | 0.72–1.84 | t = 2.129 p = .034 |

.302), with a significant increase in the use of such interventions between the period before the pandemic and the period after the first wave (*p*< .001).

**Differences in hospitalization rates and visits to the mental health emergency department in patients who received teletherapy during the lockdown and those who did not.** Hospitalization rates two months after the lockdown did not differ between patients who received teletherapy, and those who did not (*p* = .068). However, hospitalization rates were significantly lower for the first group of patients at four (*p* = .004) and six months after the lockdown (*p* < .001).

Patients who received teletherapy made fewer visits to Mental Health Emergency Department, at two months (*p* = .030), four months (*p* = .11*NS*), and six months (*p* = .034) after lockdown, than those who did not. The results are shown at Table 3.

**Differences in hospitalization rates six months after lockdown based on the type of teletherapy.** Patients who received therapy over the phone had less hospitalizations six months later than those who did not (14.2% vs 29.6%; RR = 0.47; $\chi^2$ = 9.340; *p* = .002). Similarly, those who received videoconferencing interventions during the lockdown had less hospitalizations than those who did not (3.4% vs 22.4%; RR = 0.15; $\chi^2$ = 7.840; *p* = .005).

Table 4 shows the three models built from the multilevel logistic regression analyses. As can be seen, the factor associated with the lowest risk of having a hospitalization six months after the lockdown was receiving an intervention through videoconferencing (OR = 0.25; *p* = .012), followed by receiving an intervention over the phone (OR = 0.50; p = .003). These findings were maintained when including potential confounders, with lower risk for hospitalizations associated with receiving in-person interventions (OR = 0.56; *p* = .003), being a woman (OR = 1,50; p = .04) and being older (OR = 1.01; *p* = .049), whereas diagnosis and treatment adherence were not significantly associated with more hospitalizations (model 3).

## Discussion

### Key results and interpretation

We performed a retrospective analysis to study the impact of the COVID-19 pandemic on the psychotherapeutic care received by patients with SMI, examining the use of teletherapy

**Table 4. Multilevel logistic regression models predicting hospitalizations 6 months after the lockdown.**

| Factor | Model 1: Interventions | | | | Model 2: Interventions +Sociodemographic Factors | | | | Model 3: Interventions +Sociodemographic +Clinical Factors | | | | |
|---|---|---|---|---|---|---|---|---|---|---|---|---|---|
| | B (SE) | β | p | 95% CI | B (SE) | β | p | 95%CI | B (SE) | β | p | 95%CI | Wald |
| **Videoconferencing**[a] | -1.58 (0.55) | 0.25 | .012 | 0.09 to 0.74 | -1.45 (0.55) | 0.23 | .008 | -2.5 to -0.37 | -1.32 (0.55) | 0.27 | .002 | 0.09 to 0.79 | -2.40 |
| **Over the phone**[b] | -0.69 (0.23) | 0.50 | .003 | 0.31 to 0.78 | -0.71 (0.23) | 0.50 | .002 | -1.16 to -0.25 | -0.75 (0.24) | 0.47 | .002 | 0.30 to 0.75 | -3.13 |
| **In-person**[c] | -0.57 (0.19) | 0.56 | .003 | 0.39 to 0.83 | -0.59 (0.19) | 0.56 | .003 | -0.97 to-0.20 | -0.62 (0.2) | 0.54 | .001 | 0.36 to 0.79 | -3.10 |
| **Being a woman**[d] | | | | | 0.45 (0,18) | 1.56 | .016 | 0.08 to 0.80 | 0.41 (0.20) | 1.50 | .040 | 1.02 to 2.21 | 2.05 |
| **Age (in years)** | | | | | 0.01 (0.01) | 1.01 | .83 | -0.002 to 0.003 | 0.02 (0.01) | 1.02 | .049 | 1.00 to 1.03 | 2.00 |
| **Bipolar Disorder**[e] | | | | | | | | | 0.04 (0.25) | 1.04 | .868 | 0.64 to 1.69 | |
| **Personality Disorder**[e] | | | | | | | | | 0.51 (0.30) | 1.66 | .092 | 0.92 to 3.00 | |
| **Depressive Disorder**[e] | | | | | | | | | -0.41 (0.38) | 0.27 | .661 | 0.32 to 1.06 | |
| **Anxiety/Other Disorders**[e] | | | | | | | | | -0.48 (0.28) | 0.62 | .081 | 0.36 to 1.06 | |
| **Treatment Adherence**[f] | | | | | | | | | -0.36 (0.24) | 0.69 | .129 | 0.43 to 1.11 | |

Reference groups:

[a,b,c] Not having received this intervention;

[d]: Being a man;

[e]: Psychosis;

[f]: No/Partial Adherence

alternatives applied when in-person interventions were not feasible. We also examined the potential associations between different types of teletherapy and SMI patients' hospitalization rates and visits to emergency departments in the medium term.

Our findings suggested that individual in-person interventions were significantly reduced during the lockdown, while group therapy nearly disappeared. On the other hand, teletherapy over the phone was the type most frequently used during the lockdown, perhaps because this was the only type of teletherapy intervention used before the pandemic. Teletherapy via video-conferencing, especially in group format, was introduced during the lockdown, but the percentage of patients who received it was lower in comparison with the over the phone modality. The preference for over the phone interventions may be explained by the scarcity of online platforms available within the Spanish public healthcare system [40], and perhaps also by certain limitations associated with videoconferencing, such as the difficulty of its use [41], poor training [42], or the perception that videoconferencing may increase the clinical workload [43]. Also, previous findings indicate that patients with SMI prefer using the telephone rather than videoconferencing, as they believe it reduces the intensity of the sessions [44]. Notably, to date, Spain has no established deontological framework providing guidelines for the teletherapy practice, and such practices are therefore applied in line with the traditional therapy framework principles. This has created major loopholes [45], especially concerning the information confidentiality and privacy [14, 15].

In our study, the use of the in-person modality returned to previous levels after the lockdown, despite a greater risk of viral transmission. This coincided with a significant decrease in the use of over the phone interventions. These findings indicate that when the restrictions were loosened and the situation was perceived as less critical, there was a clear tendency towards resuming routine intervention modes. Although there was no generalized rejection for teletherapy, professionals still seemed to prefer traditional approaches [46], particularly with high-risk patients [28]. Nonetheless, and albeit to a lesser extent, clinicians continued the implementation of group interventions by means of videoconferencing, as this modality may

have offered a good alternative for avoiding the physical gathering of various individuals within the same room, and thus preventing the spread of the virus. It therefore seems that the mental health services opted to offer their patients a range of options with which to guarantee the continuity of services, in line with the recommendations of Kopelovich et al. [47] for good practices in mental health services in times of COVID.

Importantly, for patients who received teletherapy during the lockdown, emergency visits and hospitalization rates were lower four and six months after the first wave of COVID-19. Receiving teletherapy during stressful periods, when in-person sessions are not possible, appears to be a protective factor against hospitalization, particularly in the medium-term. This concurs with previous findings suggesting that over the phone [48] and videoconferencing [41, 49] interventions have levels of clinical effectiveness, similar to those of in-person therapy. The interruption of psychotherapy in times of uncertainty and fear, may certainly have triggered a feeling of loss, helplessness, or sense of vulnerability. Patient-family cohabitation in confined spaces for a long period of time and the emotional discomfort associated with the pandemic, may also have led to tensions within the family affecting expressed emotion levels [50], a well-known risk factor for relapse [51]. In this regard, continuity of care through tele-therapy may have offset the negative effects of expressed emotions and have prevented crisis [52]. Importantly, patients that received therapy via videoconferencing seemed to have less hospitalizations six months after the first wave of the pandemic. This supports recommendations that videoconferencing should be used as a means of assuring continuity of care during the pandemic [8]. It also concurs with the conclusions of a systematic review of 65 videocon-ferencing-based psychotherapy studies that suggested videoconferencing as a feasible alternative to in-person interventions, as it is associated with positive outcomes such as treatment adherence, social functioning, and quality of life levels [53].

## Limitations and strengths

This study is the first to explore psychotherapeutic intervention alternatives for individuals with SMI, during the COVID-19 pandemic, a topic that has been largely neglected in the litera-ture. The study's observational nature, together with the participation of multiple healthcare sites allowed us to draw comprehensive conclusions regarding the clinical care provided in Spain, under the specific conditions created by the COVID-19 pandemic. The extraction of the patients' data from routine clinical practice records also ensured greater reliability of the findings, while the retrospective design with multiple follow-ups provided us with a unique opportunity to observe the associations of a variety of teletherapy types with the patients' hos-pitalization rates after the first wave of the pandemic.

However, the following limitations need to be considered. Firstly, as this is an observational study, it is not possible to establish direct causal relationships, as other confounding variables may play a role in the association between the intervention modalities and the hospitalization rates. Secondly, the study design did not allow for the comparison of teletherapy with in-per-son interventions, since, due to the special conditions created by the COVID-19 crisis in-per-son therapy was barely used during the studied period. This explains why the in-person modality was considered as a confounding factor, and not as an independent variable in the multilevel analysis. Thirdly, the retrospective nature of the study may have biased the results, despite the inclusion of objective routine data based on clinical records specifically aimed at reducing such bias. Fourthly, patients were grouped under the Serious Mental Illness condi-tion, and we did not perform a separate analysis for each diagnostic category. Diagnoses in mental health are in fact collections of symptoms and could be explained using completely dif-ferent lenses to the medical lens [54]. Therefore, including all patients with SMI in our study

allowed for an overall and realistic perspective of the routine clinical practice during the first wave of the pandemic. It should be noted that, the multilevel findings suggested that the probability of having a hospitalization six months after the first wave did not seem to vary between patients with different diagnoses. Similarly, previous studies have reported associations between sociodemographic and clinical factors such as sex, age, medication adherence, and hospitalizations [55, 56]. We also included those factors as confounding variables in our multilevel analyses, to control for their potential contribution. Although it lay beyond the scope of this study to focus on such associations, future research should conduct a more in-depth examination of each diagnostic category and of the role played by sociodemographic factors in the context of teletherapy. Lastly, information on theoretical approaches of the specific interventions, was not taken into account for our analyses. This was beyond the scope of the present study, which focused exclusively on the use of teletherapy as a valid alternative in the unique restrictive context of the pandemic.

### Future research perspectives

Future research into the topic should compare the implementation of teletherapy in different diagnostic categories of SMI, and also examine its impact on psychosocial outcomes, such as functioning and recovery. Extending the follow-up period would also make it possible to observe any potential changes in the use of teletherapy in the long run, while a detailed description of the specific interventions performed within each therapeutic mode would provide useful information regarding the feasibility of their telematic alternatives. Finally, the application of experimental designs such as randomized control trials would throw light on the effectiveness of teletherapy, beyond the context of the pandemic, as current results are inconsistent [57]. It is worth noting that, to date, no official guidelines exist in Spain for the use of telematic means in healthcare, making this a challenging task in this context [58].

### Conclusions

In conclusion, the COVID-19 pandemic seems to have brought changes in the practice of psychological interventions, as in-person interventions decreased, and the use of teletherapy increased, providing an opportunity to explore its applicability with individuals with SMI. This study discusses the implications of the pandemic on the provision of care for individuals with SMI, and opens the door to further research on the effectiveness of teletherapy beyond the pandemic context. Our findings suggest that teletherapy can indeed serve as a valid alternative for protecting patients with SMI against hospitalizations, especially under circumstances where in-person interventions are not feasible.

### Supporting information

**S1 File. G-power_analysis.**
(PDF)

**S1 Checklist. STROBE-checklist.**
(PDF)

### Acknowledgments

The authors wish to acknowledge the collaboration of the day hospitals section of the Spanish Neuropsychiatry Association.

## Author Contributions

**Conceptualization:** Antonio José Sánchez-Guarnido, Beatriz Machado Urquiza, Maria del Mar Soler Sánchez, Carmen Masferrer, Francisca Perles, Eleni Petkari.

**Data curation:** Antonio José Sánchez-Guarnido.

**Formal analysis:** Antonio José Sánchez-Guarnido.

**Funding acquisition:** Antonio José Sánchez-Guarnido, Maria del Mar Soler Sánchez, Carmen Masferrer, Francisca Perles.

**Investigation:** Antonio José Sánchez-Guarnido, Beatriz Machado Urquiza, Maria del Mar Soler Sánchez.

**Methodology:** Antonio José Sánchez-Guarnido.

**Supervision:** Eleni Petkari.

**Writing – original draft:** Antonio José Sánchez-Guarnido, Beatriz Machado Urquiza, Maria del Mar Soler Sánchez.

**Writing – review & editing:** Antonio José Sánchez-Guarnido, Beatriz Machado Urquiza, Maria del Mar Soler Sánchez, Carmen Masferrer, Francisca Perles, Eleni Petkari.

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
