## [Decision Letter · Decision Letter 0]

11 Feb 2022

PONE-D-21-24939Teletherapy and hospitalizations in patients with serious mental illness during the COVID-19 pandemic: A retrospective multicenter studyPLOS ONE

Dear Dr. Petkari,

Thank you for submitting your manuscript to PLOS ONE. After careful consideration, we feel that it has merit but does not fully meet PLOS ONE’s publication criteria as it currently stands. Therefore, we invite you to submit a revised version of the manuscript that addresses the points raised during the review process.

We look forward to receiving your revised manuscript.

Kind regards,

Xenia Gonda

Academic Editor

PLOS ONE

Journal Requirements:

Reviewers' comments:

Reviewer's Responses to Questions

**Comments to the Author**

1. Is the manuscript technically sound, and do the data support the conclusions?

Reviewer #1: Yes

Reviewer #2: Yes

Reviewer #3: Yes

2. Has the statistical analysis been performed appropriately and rigorously? 

Reviewer #1: No

Reviewer #2: Yes

Reviewer #3: I Don't Know

3. Have the authors made all data underlying the findings in their manuscript fully available?

Reviewer #1: No

Reviewer #2: Yes

Reviewer #3: Yes

4. Is the manuscript presented in an intelligible fashion and written in standard English?

Reviewer #1: Yes

Reviewer #2: Yes

Reviewer #3: No

5. Review Comments to the Author

Reviewer #1: Thanks for giving me the opportunity to review this paper. The paper covers an important and current topic of remotely delivered mental health services during COVID-19 pandemic. I have few comments which may need to be addressed by the authors.

1) I think it would be a good idea if the authors provide more details for how the ‘treatment adherence’ was measured in the methodology section.

2) I wonder if the authors can explore further how the % of hospitalization was calculated? Was it calculated against the individual’s history of admission or to all patients in the current sample.

3) I am not sure if McNamar test was the appropriate test to examine the changes over the three periods of time. This test examines each couple of periods individually, so the error may be amplified over the three comparisons. I think Friedman test would be more fitting here.

4) Similarly, table three demonstrates individual lines of analyses (chi square or t test for each line), which I find difficult to interpret. It could be more comprehensible to apply one chi square analysis for one table (3x2) and the same for t test, instead ANOVA test would be used.

5) It is not clear why authors included these specific factors in the hierarchal regression model (specific sociodemographic or clinical). I may suggest adding some details in the introduction section to justify the work.

6) From table 4, I would appreciate if the authors could provide the Wald value for the significant predictors to identify the strongest predictor, given that all interventions were significant, including the In-person intervention. Particularly when the CI of the Videoconference intervention is quite high.

Reviewer #2: The article reflects a topic of very current interest. They carry out an adequate bibliographic review of the topic. The methodology is correct and they meet the ethical requirements for this issue. In the discussion the critical aspects of the work are reviewed in a very appropriate way. It is important that the deficit aspects are well specified in the limitations section.

Reviewer #3: Peer review PONE

Thank you for the opportunity to review this manuscript entitled “Teletherapy and hospitalizations in patients with serious mental illness during the COVID-19 pandemic: A retrospective multicenter study”

Please find my comments below.

Major Comments

1) Please include a clear definition of each of the modalities you are talking about in the introduction and use consistent terms throughout (i.e. what is involved in “videoconferencing”? Does it have to involve more than 2 parties? Did this vary at all for the people involved in the study/across hospitals?)

2) In line 146 you state: “Serious Mental Illness was defined based on the ICD diagnosis, as well as on the intensity of the required care provision”. But you do not specify which diagnoses specifically were considered SMI. “SMI” is not a diagnosis itself. Your sample includes people with anxiety disorders – this is usually not categorised as a SMI. In addition to this clarification regarding the definition of SMI, some justification and discussion of the heterogeneity of diagnoses included in the study is needed. Why not focus on one diagnosis? How do you think the heterogeneity of diagnoses might be impacting the results? Please add this to the discussion, explaining the reasoning and the possible impact of this inclusion of different diagnoses.

3) More information is needed on the G*Power analysis: what effect size were you using? How was that justified/what was it based on? The full calculation should be found in the appendix.

4) IS there a reason that the results not reported according to a standardised checklist? Please see a list of checklists and arrange the results in line with a standardised approach, including the checklist in the appendix: https://www.strobe-statement.org/checklists/

5) It would be useful to a have a few lines on the mental health system in Spain where the study took place. It seems from the method section that there are specific Accident and Emergency units for mental health. Is this the case? A small introduction to the context would help the reader – this could be added to the method or to the introduction.

6) More explanation and justification is needed in the statistical analysis section. For example, why was McNemar’s test used?

7) Lines 249-250: “Statistically significant differences were also found in the use of phone interventions before and after the first wave (p < .001).” Please give more details: what was this statistically significant difference? What was bigger/smaller?

8) Line 288-289: please revise, it is not clear what the sense of the sentence is.

9) The language in the discussion section “Principal Results and Comparison with Prior Work” of the finding that having more videoconferencing was associated with lower hospitalisation rates suggests causality. However, the study is observational (which is reflected in the limitations section). This finding, though striking and important, should be described with great care.

Minor Comments

10) The methods of the abstract is hard to follow – please break into 2 sentences.

11) The aims of the study are expressed in a way that is difficult to understand. A suggested alternative follows (to replace lines 131-139). “The present study aimed to: a) explore the types of care offered to people with SMI during the first COVID-19 wave, the alternatives used when in person interventions were not possible, and the changes in the modality of interventions used over time (before the first wave, during lockdown, and after the first wave); b) examine whether receiving teletherapy, compared to not receiving teletherapy during the lockdown was associated with the frequency of visits to the emergency department and of hospital admissions, two, four, and six months after the lockdown; and c) examine if different teletherapy types (modalities?) are associated with hospitalization rates six months after the lockdown.”

12) Line 329: “telematic intervention”: what does this mean? Using consistent words/terms throughout and not introducing new terms in the discussion section will make it easier for the reader to follow.

13) Line 329: “the percentage of patients that received it was lower” lower than what? Please clarify.

14) Please have this carefully read by a native English speaker. Terms like “originated” (line 66), “smoothened” (line 367) are not incorrect but very unusual. Throughout “videoconference” should be changed in most cases to “videoconferencing”, or another new term that makes it clearer what you mean (“video call”?).

15) Lines 59-60: please rephrase, people being “characterised” by social networks is not quite correct. Perhaps change to say: “Social distancing practices may pose a great negative impact on individuals with pychotic disorders, as they are known to have small and low quality social networks”

16) line 352: please replace the number 45 with words.

17) Line 382: “recompilation” what does this mean?

6. PLOS authors have the option to publish the peer review history of their article (what does this mean?). If published, this will include your full peer review and any attached files.

Reviewer #1: **Yes: **Reham Shalaby

Reviewer #2: **Yes: **Cristina Romero-Lopez-Alberca

Reviewer #3: No

---

## [Author Response · Author response to Decision Letter 0]

10 Mar 2022

Manuscript ID PONE-D-21-24939

Response to the Reviewers’ comments

Reviewer #1: Thanks for giving me the opportunity to review this paper. The paper covers an important and current topic of remotely delivered mental health services during COVID-19 pandemic. I have few comments which may need to be addressed by the authors.

1) I think it would be a good idea if the authors provide more details for how the ‘treatment adherence’ was measured in the methodology section.

___This is now specified at the manuscript, Section “Variables and Data sources/measurement, page…, as follows:

“Treatment adherence was determined through the Medication Possession Ratio, defined as the proportion of time when medication supply is available [38]”

Andrade, S. E., Kahler, K. H., Frech, F., & Chan, K. A. (2006). Methods for evaluation of medication adherence and persistence using automated databases. Pharmacoepidemiology and drug safety, 15(8), 565-574.

2) I wonder if the authors can explore further how the % of hospitalization was calculated? Was it calculated against the individual’s history of admission or to all patients in the current sample

___The percentage of hospitalizations is calculated against the study sample. 

We have now added this to the Methods section (Outcome Variables, page 10)

3) I am not sure if McNamar test was the appropriate test to examine the changes over the three periods of time. This test examines each couple of periods individually, so the error may be amplified over the three comparisons. I think Friedman test would be more fitting here.

—-We thank the Reviewer for the suggestion. Given that our data are expressed in proportions, to perform comparisons between the different time points we have calculated a series of Cochran´s Q tests. Results can be seen at Table 2.

___We have kept the McNemar results displaying the between time points differences, for the interest of the Reader

—--We have clarified this to the Statistical methods section, page 11, as follows:

“we used a series of Cochrane’s tests to determine whether there were statistically significant differences in the proportion of patients making use of the different intervention types over the three time points (before, during, and after the lockdown). We also performed pairwise comparisons through a series of McNemar's tests, to check whether the proportion of patients using the interventions was sustained or varied from one time point to another (before, during and after the lockdown)” 

4) Similarly, table three demonstrates individual lines of analyses (chi square or t test for each line), which I find difficult to interpret. It could be more comprehensible to apply one chi square analysis for one table (3x2) and the same for t test, instead ANOVA test would be used.

___We thank the Reviewer for the suggestion. However, the individual lines of analyses in Table 3 represent independent comparisons between people that have received teletherapy and those that have not (two groups). Therefore, we performed three chi-square tests to compare the two groups in terms of Hospitalization rates (for each of the three different time points). Similarly, we performed three t-test analyses to compare the two groups in terms of Emergency visits. 

5) It is not clear why authors included these specific factors in the hierarchal regression model (specific sociodemographic or clinical). I may suggest adding some details in the introduction section to justify the work.

__The sociodemographic and clinical variables added to the model were chosen based on previous evidence suggesting such variables as potential risk factors for relapse. It was beyond the scope of the current study to further explore their potential associations with relapse, however, we considered it was important to add them in the model to control for their potential confounding effect. 

___We have now added further clarifications in the Limitations section as follows (manuscript page 23): 

“Similarly, previous studies have reported associations between sociodemographic and clinical factors such as sex, age, medication adherence, and hospitalizations [54, 55]. We also included those factors as confounding variables in our multilevel analyses, to control for their potential contribution. Although it lay beyond the scope of this study to focus on such associations, future research should conduct a more in-depth examination of each diagnostic category and of the role played by sociodemographic factors in the context of teletherapy.”

6) From table 4, I would appreciate if the authors could provide the Wald value for the significant predictors to identify the strongest predictor, given that all interventions were significant, including the In-person intervention. Particularly when the CI of the Videoconference intervention is quite high.

___We have now added the Wald value, check Table 4.

Reviewer #2: The article reflects a topic of very current interest. They carry out an adequate bibliographic review of the topic. The methodology is correct and they meet the ethical requirements for this issue. In the discussion the critical aspects of the work are reviewed in a very appropriate way. It is important that the deficit aspects are well specified in the limitations section.

___Thank you for your comments and the positive feedback to our manuscript

Reviewer #3: Peer review PONE

Thank you for the opportunity to review this manuscript entitled “Teletherapy and hospitalizations in patients with serious mental illness during the COVID-19 pandemic: A retrospective multicenter study”

Please find my comments below.

Major Comments

1) Please include a clear definition of each of the modalities you are talking about in the introduction and use consistent terms throughout (i.e. what is involved in “videoconferencing”? Does it have to involve more than 2 parties? Did this vary at all for the people involved in the study/across hospitals?)

___We have now provided more complete definitions for each of the teletherapy modalities (Introduction page 5). 

___In the methods section, we have provided more details regarding the specific modalities used in our study (Introduction pages 9-10) 

2) In line 146 you state: “Serious Mental Illness was defined based on the ICD diagnosis, as well as on the intensity of the required care provision”. But you do not specify which diagnoses specifically were considered SMI. “SMI” is not a diagnosis itself. Your sample includes people with anxiety disorders – this is usually not categorised as a SMI. In addition to this clarification regarding the definition of SMI, some justification and discussion of the heterogeneity of diagnoses included in the study is needed. Why not focus on one diagnosis? How do you think the heterogeneity of diagnoses might be impacting the results? Please add this to the discussion, explaining the reasoning and the possible impact of this inclusion of different diagnoses.

___We have adopted the National Institute of Mental Health definition for Severe Mental Illness, as such perspective allows for the classification of patients not only based on their Diagnosis, but also on the severity of their condition in terms of functional impairment that substantially interferes with the person's everyday life, and thus calls for intensive care. Besides Psychosis and Affective disorders, anxiety disorders, eating disorders, and personality disorders are also considered Severe Mental Illnesses when a) the degree of functional impairment is severe (Evans et al., 2016; Wing et al., 2004) and b) there is a need for an interdisciplinary treatment approach (Moreno et al., 2020).

In the Spanish Healthcare system, such an approach is offered by the Community Mental Health Hospitals network, that is destined to patients with major complexity in terms of functional impairment and treatment needs. Therefore, our sample consisted of all the patients that attended any of the Community Mental health Hospitals that took part in the study, as those patients were fulfilling the above-mentioned conditions, and therefore had the need of attending such services.

Evans TS, Berkman N, Brown C, Gaynes B, Weber RP. Disparities within serious mental illness. Agency for Healthcare Briefs and Quality, Rockville, USA. 2016; 25.

Moreno, M. J., Jaén, M. J., Lillo, R., Guija, J. A., & Medina, A. Severe Mental Illness: Psychiatry and Law. Spanish Federation of Psychiatry and Mental Health, Madrid, Spain. 2020 (book in Spanish)

Wing JK. Severe Mental Illness. In: Stevens a (ed). Health Care Needs Assessment: The Epidemiologically Based Needs. Assessment Reviews, vol 2. Oxford-San Francisco. Radcliff Publishing. 2004; 159-237.

___We agree with this Reviewer that the inclusion of all patients with SMI without distinguishing among different diagnoses may influence our results, therefore we included Diagnosis as a confounding variable to our multilevel analyses. The findings suggested that Diagnosis did not seem to be associated with hospitalizations six months after the first wave (check Methods and Results sections). 

____However, we still acknowledge that Diagnostic heterogeneity may play a role, and added this to the Limitations of our study:

“Fourthly, patients were grouped under the Serious Mental Illness condition, and we did not perform a separate analysis for each diagnostic category. Including all patients with SMI in our study allowed for an overall and realistic perspective of the routine clinical practice during the first wave of the pandemic. It should be noted that, the multilevel findings suggested that the probability of having a hospitalization six months after the first wave did not seem to vary between patients with different diagnoses…Although it lay beyond the scope of this study to focus on such associations, future research should conduct a more in-depth examination of each diagnostic category and of the role played by sociodemographic factors in the context of teletherapy”

3) More information is needed on the G*Power analysis: what effect size were you using? How was that justified/what was it based on? The full calculation should be found in the appendix.

___More details were added to the Setting and Participants section, page 9 as follows:

“To calculate the sample size, we considered a 20% relapse rate for the group that received teletherapy and 30% relapse rate for the group that did not, with a potential sample loss of 15%, a Confidence Interval of 95% and 80% statistical power (see S1 for details).” 

___We have added the G-power calculation at the appendix (S1), using the corresponding formula provided at: https://www.fisterra.com/mbe/investiga/9muestras/9muestras2.asp

4) IS there a reason that the results not reported according to a standardised checklist? Please see a list of checklists and arrange the results in line with a standardised approach, including the checklist in the appendix: https://www.strobe-statement.org/checklists/

___As per the recommendations of this Reviewer, we have adapted the manuscript following the STROBE checklist approach. 

___We include the checklist as a supplementary file (S2).

5) It would be useful to a have a few lines on the mental health system in Spain where the study took place. It seems from the method section that there are specific Accident and Emergency units for mental health. Is this the case? A small introduction to the context would help the reader – this could be added to the method or to the introduction.

___We have now added a paragraph briefly explaining the Spanish mental healthcare system. See Methods, Settings and Participant Recruitment section, page 8

6) More explanation and justification is needed in the statistical analysis section. For example, why was McNemar’s test used?

___Please see answer to Comment 3 Reviewer 1

7) Lines 249-250: “Statistically significant differences were also found in the use of phone interventions before and after the first wave (p < .001).” Please give more details: what was this statistically significant difference? What was bigger/smaller?

___We have modified the sentence to provide clarity as follows (page 14)

 “However, this percentage was still significantly higher than before the lockdown (p < .001).” 

8) Line 288-289: please revise, it is not clear what the sense of the sentence is.

___We have now revised the sentence to provide clarity.

9) The language in the discussion section “Principal Results and Comparison with Prior Work” of the finding that having more videoconferencing was associated with lower hospitalisation rates suggests causality. However, the study is observational (which is reflected in the limitations section). This finding, though striking and important, should be described with great care.

___We have now modified the sentence to provide clarity and avoid causality conclusions, as follows (page 21):

“Importantly, patients that received therapy through videoconferencing seemed to have less hospitalizations six months after the first wave of the pandemic…”

Minor Comments

10) The methods of the abstract is hard to follow – please break into 2 sentences.

__We have now restructured the methods section of the abstract, to provide clarity

11) The aims of the study are expressed in a way that is difficult to understand. A suggested alternative follows (to replace lines 131-139). “The present study aimed to: a) explore the types of care offered to people with SMI during the first COVID-19 wave, the alternatives used when in person interventions were not possible, and the changes in the modality of interventions used over time (before the first wave, during lockdown, and after the first wave); b) examine whether receiving teletherapy, compared to not receiving teletherapy during the lockdown was associated with the frequency of visits to the emergency department and of hospital admissions, two, four, and six months after the lockdown; and c) examine if different teletherapy types (modalities?) are associated with hospitalization rates six months after the lockdown.”

___We thank the Reviewer for this suggestion, which we used to replace the original aims section. 

12) Line 329: “telematic intervention”: what does this mean? Using consistent words/terms throughout and not introducing new terms in the discussion section will make it easier for the reader to follow.

___Please check page 5, where telematic psychological interventions are defined as follows, based on the literature:

“Alternatives to in-person care were in fact implemented with the aid of teletherapy, un umbrella term for what are known as telematic psychological interventions (telepsychotherapy and e-therapy), Teletherapy is based on information and communications technology (computer-based Internet tools, mobile and land telephone calls, emails, fax, text messages, and videoconferencing consisting of patient-clinician communication through video consultations) [12].”

___We have also modified the term across the text replacing it with the term teletherapy when feasible, and consistent with the meaning.

13) Line 329: “the percentage of patients that received it was lower” lower than what? Please clarify.

___We have now specified that the percentage of patients that received teletherapy through videoconference was lower as compared to those receiving the over the phone modality

14) Please have this carefully read by a native English speaker. Terms like “originated” (line 66), “smoothened” (line 367) are not incorrect but very unusual. Throughout “videoconference” should be changed in most cases to “videoconferencing”, or another new term that makes it clearer what you mean (“video call”?).

___We have now replaced videoconference with videoconferencing, following the suggestion of this Reviewer.

___ We have also replaced originated with caused by. smoothened with offset. 

—-We also had a native English speaker proofreading our text and taken care of potentially unclear sentences. 

15) Lines 59-60: please rephrase, people being “characterised” by social networks is not quite correct. Perhaps change to say: “Social distancing practices may pose a great negative impact on individuals with pychotic disorders, as they are known to have small and low quality social networks”

___We have replaced the sentence as per the instructions of this Reviewer

16) line 352: please replace the number 45 with words.

___We have now added the first author´s name for this citation

17) Line 382: “recompilation” what does this mean?

___We have now replaced recompilation with “extraction” to provide clarity.

We would like to thank the Reviewers for their valuable and helpful suggestions that helped us improve our manuscript.

---

## [Decision Letter · Decision Letter 1]

29 Mar 2022

PONE-D-21-24939R1Teletherapy and hospitalizations in patients with serious mental illness during the COVID-19 pandemic: A retrospective multicenter studyPLOS ONE

Dear Dr. Petkari,

Thank you for submitting your manuscript to PLOS ONE. After careful consideration, we feel that it has merit but does not fully meet PLOS ONE’s publication criteria as it currently stands. Therefore, we invite you to submit a revised version of the manuscript that addresses the points raised during the review process.

We look forward to receiving your revised manuscript.

Kind regards,

Xenia Gonda

Academic Editor

PLOS ONE

Journal Requirements:

Additional Editor Comments:

As you will see, the reviewers found that you have addressed all your comments which improved this already excellent paper. Before your paper is ready to be accepted, please address the minor comments raised by Reviewer 1.

Reviewers' comments:

Reviewer's Responses to Questions

**Comments to the Author**

1. If the authors have adequately addressed your comments raised in a previous round of review and you feel that this manuscript is now acceptable for publication, you may indicate that here to bypass the “Comments to the Author” section, enter your conflict of interest statement in the “Confidential to Editor” section, and submit your "Accept" recommendation.

Reviewer #1: All comments have been addressed

Reviewer #3: All comments have been addressed

2. Is the manuscript technically sound, and do the data support the conclusions?

Reviewer #1: Yes

Reviewer #3: Yes

3. Has the statistical analysis been performed appropriately and rigorously? 

Reviewer #1: Yes

Reviewer #3: I Don't Know

4. Have the authors made all data underlying the findings in their manuscript fully available?

Reviewer #1: (No Response)

Reviewer #3: Yes

5. Is the manuscript presented in an intelligible fashion and written in standard English?

Reviewer #1: (No Response)

Reviewer #3: Yes

6. Review Comments to the Author

Reviewer #1: I would like to thank the authors for the prompt reply to the raised points.

Only few concerns are still there:

1) Thanks for using Cochrane’s test, I think, therefore the authors would need to report McNemar’s test with manually Bonferroni correction, rather than McNemar’s test only.

2) Table 3 seems quite confusing to me. The title refers to six months after the first wave, while in the table there are two months after the lock down. Similarly, in narration, the two terms were used interchangeably “Hospitalization rates two months after the first wave did not differ between patients who received …”, please consider revision

3) I may disagree with “… with slight risks associated with receiving in-person interventions (OR=0.56; p=.03)”, this seems to me not a risk, however in-person interventions seems going in the same direction as videoconferencing and interventions over the phone, with a lower risk for hospitalization.

4) A typo in “(p = .11NS)” and in Table 4 “-0,69 (0.23)”

Reviewer #3: Peer review PONE

Dear Authors,

Thank you for the thorough responses to my queries. I think a final point on the diagnosis question, although you have addressed this very thoroughly with the changes to the method and discussion, is maybe to hint at the fact that diagnoses in mental health are all collections of symptoms, or syndromes, and could be explained using completely different lenses to the medical lens. This, I think, would aid in making the argument for why it was justified, and better, that you did not stick very strictly to one country’s narrow diagnostic category, from a single point in time, but rather took a more inclusive view, that reflects the uncertainties around the nature and aetiology of mental illness and our ever-evolving understandings.

Some references to include in making this point (a sentence in the discussion or methods would suffice I believe):

Bhui, K., & Priebe, S. (2006). Assessing explanatory models for common mental disorders. The Journal of clinical psychiatry, 67(6), 1441.

Conneely, M., Higgs, P., & Moncrieff, J. (2021). Medicalising the moral: the case of depression as revealed in internet blogs. Social Theory & Health, 19(4), 380-398.

There are a few minor changes that need correcting that should be picked up in a thorough proof-read, a few are picked up below:

1. Repetition of the word “challenging” in the concluding paragraph of the abstract.

2. “Less” should be “fewer” in the last line of the Results section of the abstract.

3. First paragraph of the discussion, last sentence, there is an unnecessary “of”

4. Discussion: “crise” � “crisis”

Congratulations on this impressive piece of work!

7. PLOS authors have the option to publish the peer review history of their article (what does this mean?). If published, this will include your full peer review and any attached files.

Reviewer #1: **Yes: **Reham Shalaby

Reviewer #3: No

---

## [Author Response · Author response to Decision Letter 1]

1 Apr 2022

Manuscript ID PONE-D-21-24939-R1

Response to the Reviewers’ comments

Reviewer #1: I would like to thank the authors for the prompt reply to the raised points.

Only few concerns are still there:

1) Thanks for using Cochrane’s test, I think, therefore the authors would need to report McNemar’s test with manually Bonferroni correction, rather than McNemar’s test only.

_We added a sentence in the Statistical Methods section specifying the Bonferroni correction applied (page 11)

_ To reflect the applied correction to the results, we modified the corresponding Results section (page 14)

2) Table 3 seems quite confusing to me. The title refers to six months after the first wave, while in the table there are two months after the lock down. Similarly, in narration, the two terms were used interchangeably “Hospitalization rates two months after the first wave did not differ between patients who received …”, please consider revision

___We have now revised the Table title and the text to provide consistency

3) I may disagree with “… with slight risks associated with receiving in-person interventions (OR=0.56; p=.03)”, this seems to me not a risk, however in-person interventions seems going in the same direction as videoconferencing and interventions over the phone, with a lower risk for hospitalization.

___We have now modified this sentence to provide clarity, as follows: “These findings were maintained when including potential confounders, with lower risk for hospitalizations associated with receiving in-person interventions(…)”

4) A typo in “(p = .11NS)” and in Table 4 “-0,69 (0.23)”

__Thank you for the suggestion, these typos are corrected

Reviewer #3: Peer review PONE

Dear Authors,

Thank you for the thorough responses to my queries. I think a final point on the diagnosis question, although you have addressed this very thoroughly with the changes to the method and discussion, is maybe to hint at the fact that diagnoses in mental health are all collections of symptoms, or syndromes, and could be explained using completely different lenses to the medical lens. This, I think, would aid in making the argument for why it was justified, and better, that you did not stick very strictly to one country’s narrow diagnostic category, from a single point in time, but rather took a more inclusive view, that reflects the uncertainties around the nature and aetiology of mental illness and our ever-evolving understandings.

Some references to include in making this point (a sentence in the discussion or methods would suffice I believe):

Bhui, K., & Priebe, S. (2006). Assessing explanatory models for common mental disorders. The Journal of clinical psychiatry, 67(6), 1441.

Conneely, M., Higgs, P., & Moncrieff, J. (2021). Medicalising the moral: the case of depression as revealed in internet blogs. Social Theory & Health, 19(4), 380-398.

__We thank the Reviewer for the suggestion, we have now added a sentence at the Discussion section: See Limitations, p.22

There are a few minor changes that need correcting that should be picked up in a thorough proof-read, a few are picked up below:

1. Repetition of the word “challenging” in the concluding paragraph of the abstract.

__The word is now erased

2. “Less” should be “fewer” in the last line of the Results section of the abstract.

_”Less” is replaced with “fewer”

3. First paragraph of the discussion, last sentence, there is an unnecessary “of”

__”Of” is erased

4. Discussion: “crise” � “crisis”

__This is now replaced

Once again, we thank the Reviewers for their valuable comments that brought considerable improvements to our manuscript

---

## [Editor Report · Decision Letter 2]

5 Apr 2022

Teletherapy and hospitalizations in patients with serious mental illness during the COVID-19 pandemic: A retrospective multicenter study

PONE-D-21-24939R2

Dear Dr. Petkari,

We’re pleased to inform you that your manuscript has been judged scientifically suitable for publication and will be formally accepted for publication once it meets all outstanding technical requirements.

Kind regards,

Xenia Gonda

Academic Editor

PLOS ONE
---

## [Editor Report · Acceptance letter]

8 Apr 2022

PONE-D-21-24939R2 

Teletherapy and hospitalizations in patients with serious mental illness during the COVID-19 pandemic: A retrospective multicenter study 

Dear Dr. Petkari:

I'm pleased to inform you that your manuscript has been deemed suitable for publication in PLOS ONE. Congratulations! Your manuscript is now with our production department. 

Kind regards, 

on behalf of

Dr. Xenia Gonda 

Academic Editor

PLOS ONE